# Detection of Strawberry Diseases Using a Convolutional Neural Network

**DOI:** 10.3390/plants10010031

**Published:** 2020-12-25

**Authors:** Jia-Rong Xiao, Pei-Che Chung, Hung-Yi Wu, Quoc-Hung Phan, Jer-Liang Andrew Yeh, Max Ti-Kuang Hou

**Affiliations:** 1Department of Mechanical Engineering, National United University, Miaoli 360001, Taiwan; a645411007@gmail.com (J.-R.X.); qhphan@nuu.edu.tw (Q.-H.P.); 2Miaoli District Agricultural Research and Extension Station, Miaoli 363201, Taiwan; peiche@mdais.gov.tw; 3Department of Plant Pathology and Microbiology, National Taiwan University, Taipei 106319, Taiwan; hywu0615@gmail.com; 4Department of Power Mechanical Engineering, National Tsing Hwa University, Hsinchu 300044, Taiwan; jayeh@mx.nthu.edu.tw

**Keywords:** strawberry diseases, convolution neural network, image recognition

## Abstract

The strawberry (*Fragaria* × *ananassa* Duch.) is a high-value crop with an annual cultivated area of ~500 ha in Taiwan. Over 90% of strawberry cultivation is in Miaoli County. Unfortunately, various diseases significantly decrease strawberry production. The leaf and fruit disease became an epidemic in 1986. From 2010 to 2016, anthracnose crown rot caused the loss of 30–40% of seedlings and ~20% of plants after transplanting. The automation of agriculture and image recognition techniques are indispensable for detecting strawberry diseases. We developed an image recognition technique for the detection of strawberry diseases using a convolutional neural network (CNN) model. CNN is a powerful deep learning approach that has been used to enhance image recognition. In the proposed technique, two different datasets containing the original and feature images are used for detecting the following strawberry diseases—leaf blight, gray mold, and powdery mildew. Specifically, leaf blight may affect the crown, leaf, and fruit and show different symptoms. By using the ResNet50 model with a training period of 20 epochs for 1306 feature images, the proposed CNN model achieves a classification accuracy rate of 100% for leaf blight cases affecting the crown, leaf, and fruit; 98% for gray mold cases, and 98% for powdery mildew cases. In 20 epochs, the accuracy rate of 99.60% obtained from the feature image dataset was higher than that of 1.53% obtained from the original one. This proposed model provides a simple, reliable, and cost-effective technique for detecting strawberry diseases.

## 1. Introduction

Crop pests and diseases are major problems in the agricultural industry that cause significant losses to food production. Nearly half of the world’s crops are lost due to pest infestation and disease [1]. In Miaoli County, Taiwan, the strawberry, which contributes $1.8 billion annually, is facing similar problems. Strawberries have high nutritional content, commercial value, and are a major fruit for daily consumption [2,3]. Strawberries are easily infected by several phytopathogenic fungi, bacteria, and viruses [4,5,6]. The main strawberry pathogens are as follows—*Colletotrichum siamense*, which causes anthracnose [7,8]; *Botrytis cinereal*, the causal agent of gray mold [9,10]; *Neopestalotiopsis* spp. [11], the causal agent of crown rot, fruit rot and leaf blight [7]; and other fungi that cause powdery mildew, which affects petioles [12], leaves, and fruits in a strawberry-specific manner [13]. These pathogens interfere with photosynthesis and negatively impact fruit quality, growth, and productivity. Strawberry diseases are manually identified by growers, which is laborious and time-consuming. The shrinking workforce in agricultural counties also complicates this issue, since it is harder to accurately predict disease severity over a large scale. Developing a rapid, accurate, automated technique to detect strawberry diseases is required.

Recently, deep learning artificial intelligence (AI) technology has been applied for the detection of crop diseases [14,15]. Machine learning (ML) image recognition has been used as a highly accurate and inexpensive tool for detecting crop diseases [16,17]. In published reports, the convolution neural network (CNN) is one of the most popular ML techniques for detecting crop diseases. Jeon and Rhee [18] proposed the CNN technique for plant leaf recognition using the GoogLeNet model. The proposed technique was able to detect damaged leaves with a recognition rate of >94%, even when only 30% of the leaf was damaged. Mohanty et al. [19] used CNN for detecting crop species and diseases based on a public dataset of images using GoogLeNet and AlexNet training models. Based on the color, grayscale, and leaf segmentation, the proposed model was 99.35% accurate. Selvaraj et al. [20] proposed deep transfer learning (DTL) for the detection of banana diseases and pest an the accuracy of 90%. Many deep learning AI techniques have been proposed for predicting strawberry quality. Sustika et al. [21] used five different types of deep CNN architectures, including AlexNet, MobileNet, GoogLeNet, VGGNet, and Xception for inspecting strawberry quality. These proposed techniques may be up to 95% accurate. Shin et al. [22] used supervised ML technologies to detect strawberry powdery mildew disease with an accuracy of 94.34%. These studies in [22,23,24] have provided a foundation for AI-mediated disease detection. However, CNN has not yet been used for strawberry disease detection. In this study, we used CNN techniques and a ResNet50 model to detect leaf blight (which infects the crown, leaf, and fruit), gray mold, and powdery mildew from the “Taoyuan No. 1” and “Xiang-Shui” strawberry cultivars in Miaoli County, Taiwan. The objective of this study is to apply the state-of-the-art CNN techniques for the detection of leaf blight, gray mold, and powdery mildew from the “Taoyuan No. 1” and “Xiang-Shui” strawberry cultivars in Miaoli County, Taiwan. We also compare the accuracy of results obtained from three different models namely VGG-16, GoogLeNet, and Resnet-50 to confirm the feasibility of the proposed technique for strawberry disease detection.

## 2. Results and Discussion

### 2.1. GoogLeNet Model of the Confusion Matrix

The confusion matrix shows the final detection results of the original and feature image dataset with the training period of 20 epochs of the GoogLeNet model (Figure 1). The diagonal axis number presents the correct number of classifications and the rest presented are misclassifications. The classification accuracy rate was 100% for leaf blight caused by crown rot and powdery mildew for the original image dataset (Figure 1a). The classification rate was 84% with 26 correct images for gray mold, 84% with 26 correct images for leaf blight affecting the fruit, and 90% with 28 correct images for leaf blight. For the feature image dataset, the classification accuracy rate was 100% for powdery mildew, leaf blight, and fruit rot (Figure 1b). The classification rate was 96% with 48 correct images for gray mold and 96% with 48 correct images for crown rot caused by leaf blight.

### 2.2. VGG16 Model of the Confusion Matrix

Figure 2 shows the confusion matrix of the final detection results of the original and feature image dataset with the training period of 20 epochs using the Vgg16 model. The diagonal axis number presents the correct number of classifications and the rest presented are misclassifications. For the original images dataset, the classification accuracy rate is 100% for leaf blight caused crown rot and powdery mildew (Figure 2a). The classification rate was 97% with 30 correct images for gray mold, 84% with 26 correct images for leaf blight affecting fruit, and 90% with 28 correct images for leaf blight. For the feature image dataset, the classification accuracy rate was 100% for crown rot caused by leaf blight, leaf blight, fruit rot, and gray mold as well as 96% with 48 correct images for powdery mildew (Figure 2b).

### 2.3. Resnet50 Model of the Confusion Matrix

Figure 3 shows the confusion matrix of the final detection results of the original and feature image datasets with a training period of 20 epochs for the Resnet50 model. The diagonal axis number presents the correct number of classifications and the rest presented are misclassifications. For the original images dataset, the classification accuracy rate was 100% for leaf blight-induced crown rot and leaf blight (Figure 3a). The classification rate was 97% with 30 correct images of gray mold, 84% with 26 correct images of leaf blight-induced fruit rot, and 90% with 28 correct images for leaf blight. For the feature image dataset, the classification accuracy rate was 100% for leaf blight-induced crown rot, leaf blight, and fruit rot (Figure 3b). The classification rate was 98% with 49 correct images for gray mold and 98% with 49 correct images for powdery mildew. The results also showed that for two types of leaf diseases, the system uses clear leaf patterns in the original and feature image datasets to detect without error. For the leaf blight-induced fruit rot, the classification rate increases to 100% for the feature image dataset. Thus, the trimmed image can minimize the complexity pattern and increase the accuracy of classification. Similarly, the classification rate is also improved for leaf blight causing fruit rot type from 84% by the original image dataset to 99.99% by the feature image dataset. The lower performance of the original image dataset can be explained by the complexity of the background pattern leading to the confusion of the system. Notably, this classification rate is comparable with that of the faster R-CNN multi-task learning proposed in [25,26] and higher than that of the supervised machine learning in [21,24]. In general, the proposed technique provides a potential method for five major strawberry diseases’ detection. The classification rates obtained from the feature image dataset are better than those obtained from the original image dataset with all three models. The fruit leaf blight and leaf blight diseases are detected with 100% classification rate with all three models. Furthermore, the crown leaf blight disease is detected with a 100% classification rate with only two models of VGG16 and Resnet50. The gray mold disease is detected with a 100% classification rate with only the VGG16 model. The powdery disease is detected with a 100% classification rate with only the GoogLeNet model. In the future, more research will be conducted to compare the accuracy of the results obtained by the proposed technique with those obtained from an experienced crop scout in the field. Furthermore, based on the fruitful results obtained from the proposed technique, a full automated artificial intelligent mobile app will be developed to perform disease detection in the early stage. Thus, it will control the yield losses for commercial Strawberry production in Taiwan and help millions of users in developed countries.

## 3. Materials and Methods

### 3.1. Strawberry Diseases Dataset

Images were taken for the “Taoyuan No. 1” and “Xiang-Shui” strawberry cultivars. These images were taken at a strawberry farm at Dahu Township, Miaoli County using a Sony RX10ii camera. During the shooting process, a plant pathologist identified the type of disease and marked them by nametags. Five types of strawberry disease images including leaf blight (crown rot, leaf blight, fruit rot), gray mold, and powdery mildew were collected. We choose these 5 types of disease for detection purpose because they are major diseases caused the yield lost for commercial strawberry production in Taiwan. A total of 792 images were taken and named as original images as shown in Figure 4. One original image can produce several feature images. Feature images were manually trimmed without using any background subtraction methods as shown in Figure 5. Withered areas on the leaves in the leaf blight group minimized the variation of nodule contrast.

Each of the withered leaves was cut off as a feature image. The numbers of original images and feature images are shown in Table 1. Two hundred sixty-seven feature images were trimmed from 156 original images for crown rot caused by leaf blight disease, 262 feature images were trimmed from 166 original images for leaf blight diseases, 253 feature images were trimmed from 155 original images for fruit rot caused by leaf blight disease, 250 feature images were trimmed from 157 original images for gray mold diseases, and 273 feature images were trimmed from 158 original images for powdery mildew disease. A total of 1306 images were trimmed from original images and named as feature images (Figure 6). It is noted that the healthy dataset was not collected in this study. However, in the future, the healthy dataset will be used to enhance the impact of the proposed technique.

### 3.2. Convolution Neural Network Imaging Recognition

#### 3.2.1. Convolution Neural Network Sketch

CNN is a mathematical model that can simulate brain function and neural interactive connections based on a convolution process. In CNN, the image was obtained from the input terminal and the image features are filtered out to the pooling layer through the convolution layer to sort out the new image features. The convolution and pooling are one layer. In the last layer, the final features are fully connected and identified according to the CNN algorithm. We used CNN to detect the different strawberry diseases based on the GoogLeNet, VGG16, Resnet50 algorithm. The sketch map of the CNN technique is shown in Figure 7.

Before inserting the input matrices into the neural network, the images were separated into the original and feature images. The original image dataset was used to instruct the neural network to focus on the neighborhood information and train it to extract necessary features through supervised backpropagation training. The background reduced from the feature image dataset was used to instruct the neural network to focus on the disease-like information and train the neural network to extract necessary features through supervised backpropagation training. For the output side, the neural network was trained like a pathologist making a diagnosis. This can be considered as an expert training method to convert human action into a computer algorithm.

#### 3.2.2. GoogLeNet Structure Diagram

GoogLeNet, also known as Inception V1, won the championship in the 2014 imageNet image classification competition. The input image size was 224 × 224. The network uses a sparse structure (Figure 8). The structure uses 1 × 1, 3 × 3, 5 × 5 convolution kernels and a 3 × 3 pooling layer to perform parallel operations at the same time. After the operation is completed, all the convolved feature images will overlap to generate a new feature image.

#### 3.2.3. VGG16 Structure Diagram

Vgg16 won the runner-up in the 2014 imageNet image classification competition. Vgg16 inherited the 2012 AlxeNet network design ideas and increased from eight to 16 layers on the AlexNet network. There are 13 convolutional layers (Conv), five pooling layers (Pooling), and three fully connected layers (gray parts) (Figure 9). [27]

#### 3.2.4. Resnet50 Structure Diagram

The Resnet50 model won the 2015 ILSVRC image classification championship. The Resnet50 model has a residual network layer. To solve the problem of degradation during the calculation, the residual network has a better calculation result when the roll machine layer learns new features during feature input [28]. The Resnet50 model includes five convolution layers. For instance, in the orange frame, the middle 3 × 3 convolutional layer first reduces the calculation under a dimension reduction 1 × 1 convolutional layer and then restores it under another 1 × 1 convolutional layer. This not only maintains accuracy but also reduces the amount of calculation (Figure 10).

#### 3.2.5. CNN Traning

The CNN model was coded using MATLAB. The model was run using an 80:20 ratio of 80:20 of training and testing data per training period (Figure 11). The training parameters are listed in Table 2. When performing the training, Net can only process RGB images that are 224 × 224, used enhancement color processing, converting a grayscale image to RGB. To generate a new batch, images were input into the neural network, and weights were adjusted until the image features were analyzed. Data expansion techniques to increase the image by rotation and flips were used to overcome the overfitting problem when training with a small dataset [29]. We used the CUAD^®^ the program to run the algorithm with the NVIDIA^®^RTX2070 GPU.

The average training accuracy and loss for the original and feature image datasets were calculated through three differential training periods of 20 epochs. The corresponding results of average training accuracy and loss rates were calculated for the original and feature image datasets (Figure 12 and Figure 13). The training accuracy rates for VGG-16, GoogLeNet, and Resnet-50 were 97.42%, 96.13%, and 98.06%, respectively (Figure 12). As shown in Figure 12, the training accuracy rates are 99.20%, 98.80%, and 99.60%, respectively, and VGG-16, GoogLeNet, and Resnet-50 correspond to each other. The Resnet50 model had the highest training accuracy results for the three training periods (97.42%, 96.13%, and 98.06%). The training accuracy rates for the feature data set were 99.20%, 98.80%, and 99.60% for each training period. The training accuracy rates increased with the increment of the training period of 5 to 10 epochs but subsequently decreased when the increment of the training period was changed from 15 epochs to 20 epochs. Furthermore, the accuracy of the feature image dataset was higher than that of the original image dataset. It can be explained due to the mix and complexity of the background patterns shown in the original image dataset. The system tends to be confused with several classes that results in lower performance.

## 4. Conclusions

The CNN algorithm with the Resnet50 model is proposed for detecting strawberry diseases, namely crown leaf blight, leaf blight, fruit leaf blight, gray mold, and powdery mildew, using original and feature image datasets. The training accuracy was highest at the period of 20 epochs, with 98.06% and 99.60% accuracy for the original and feature datasets, respectively. The proposed approach provides a simple technique for detecting strawberry diseases, which will be important for agricultural use.

## Figures and Tables

**Figure 1 plants-10-00031-f001:**
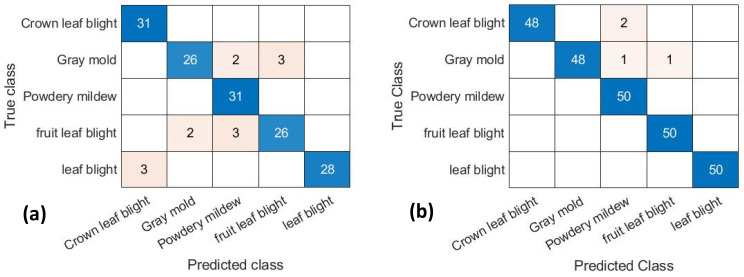
Confusion matrix with the GoogLeNet model of 20 epochs of the (**a**) original dataset, and (**b**) feature dataset.

**Figure 2 plants-10-00031-f002:**
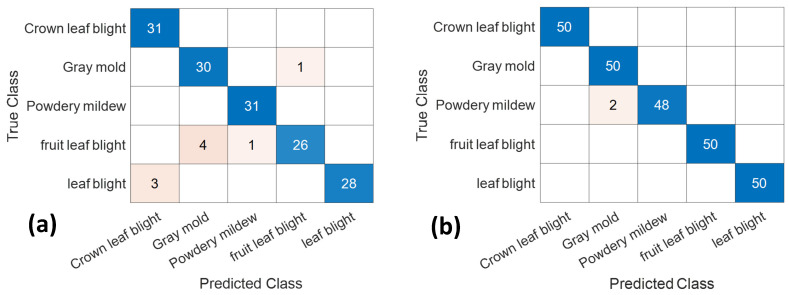
Confusion matrix with the VGG16 model of 20 epochs of the (**a**) original dataset, and (**b**) feature dataset.

**Figure 3 plants-10-00031-f003:**
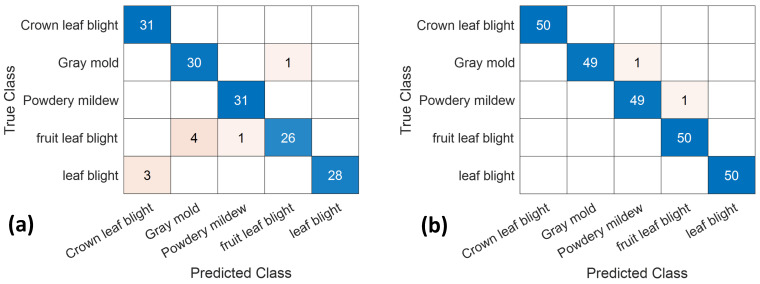
Confusion matrix with the Resnet50 model of 20 epochs of the (**a**) original dataset, and (**b**) feature dataset.

**Figure 4 plants-10-00031-f004:**
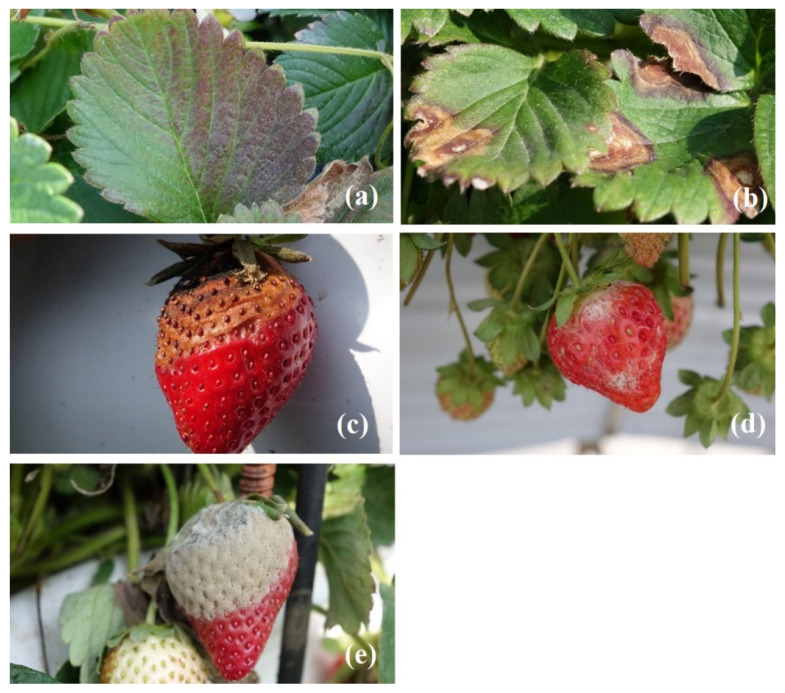
A total of 792 original images for different diseases of strawberry plant—(**a**) crown rot caused by leaf blight, (**b**) leaf blight, (**c**) fruit rot caused by leaf blight, (**d**) powdery mildew, and (**e**) gray mold.

**Figure 5 plants-10-00031-f005:**
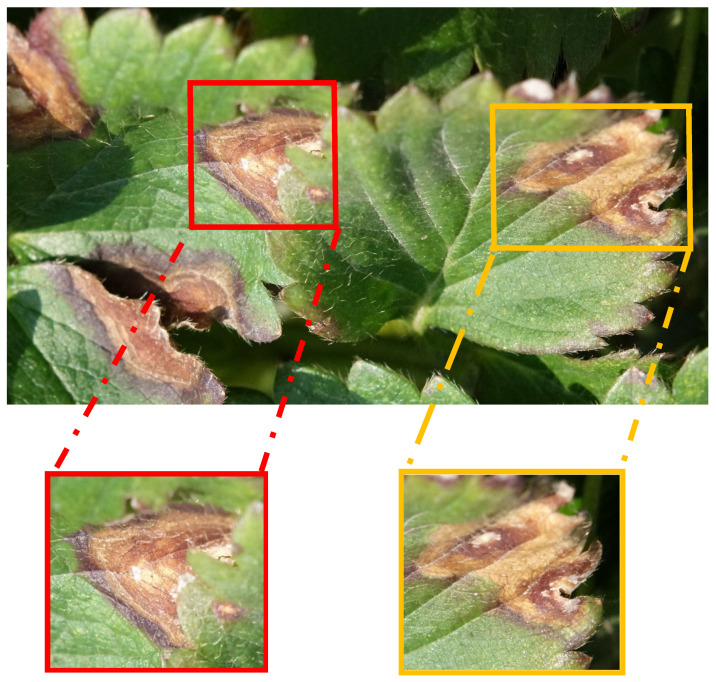
Schematic diagram of original image feature extraction.

**Figure 6 plants-10-00031-f006:**
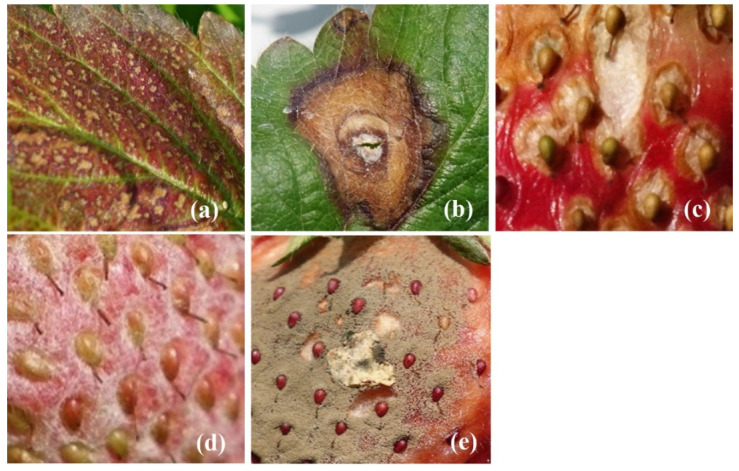
A total of 1306 feature images were trimmed from original images for different diseases of strawberry plant—(**a**) crown rot caused by leaf blight, (**b**) leaf blight, (**c**) fruit rot caused by leaf blight, (**d**) powdery mildew, and (**e**) gray mold.

**Figure 7 plants-10-00031-f007:**
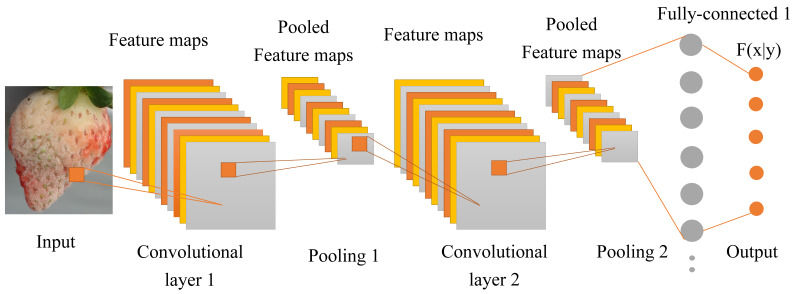
Convolution neural network for the detection of strawberry diseases.

**Figure 8 plants-10-00031-f008:**
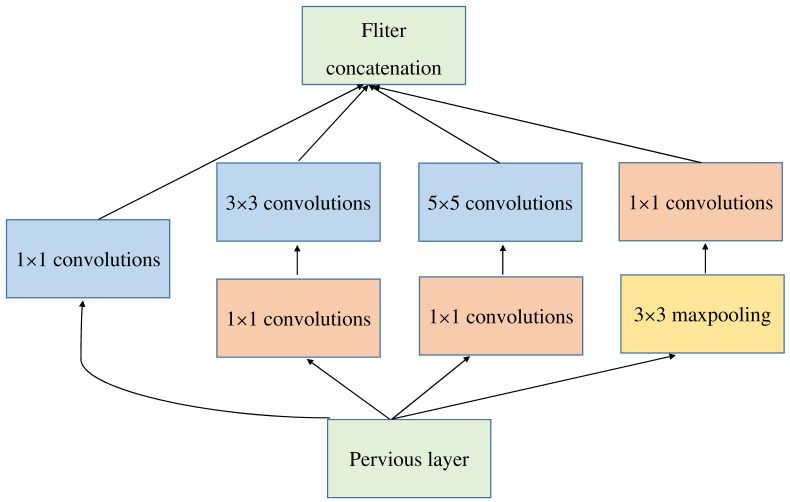
Diseases Inception Model structure diagram.

**Figure 9 plants-10-00031-f009:**
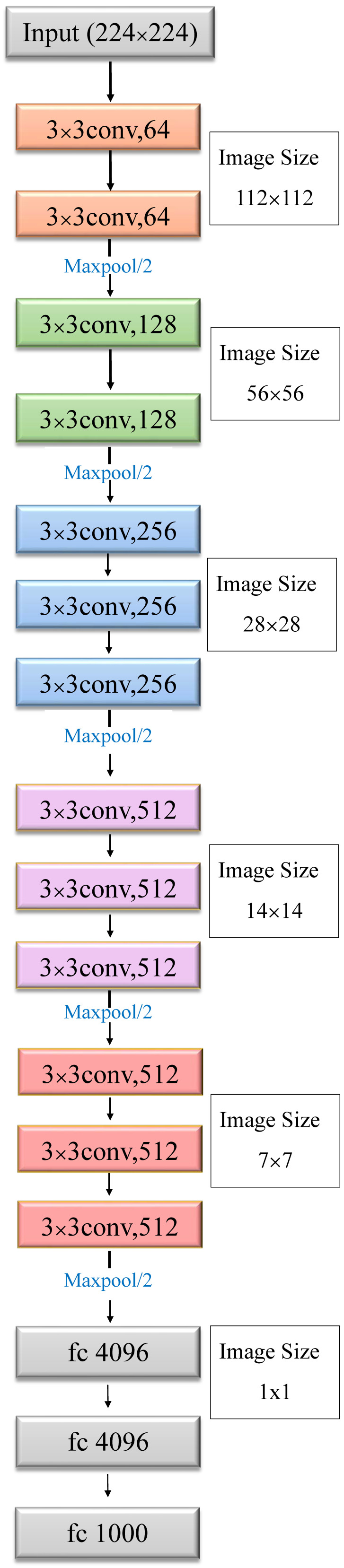
Vgg16 network architecture diagram.

**Figure 10 plants-10-00031-f010:**
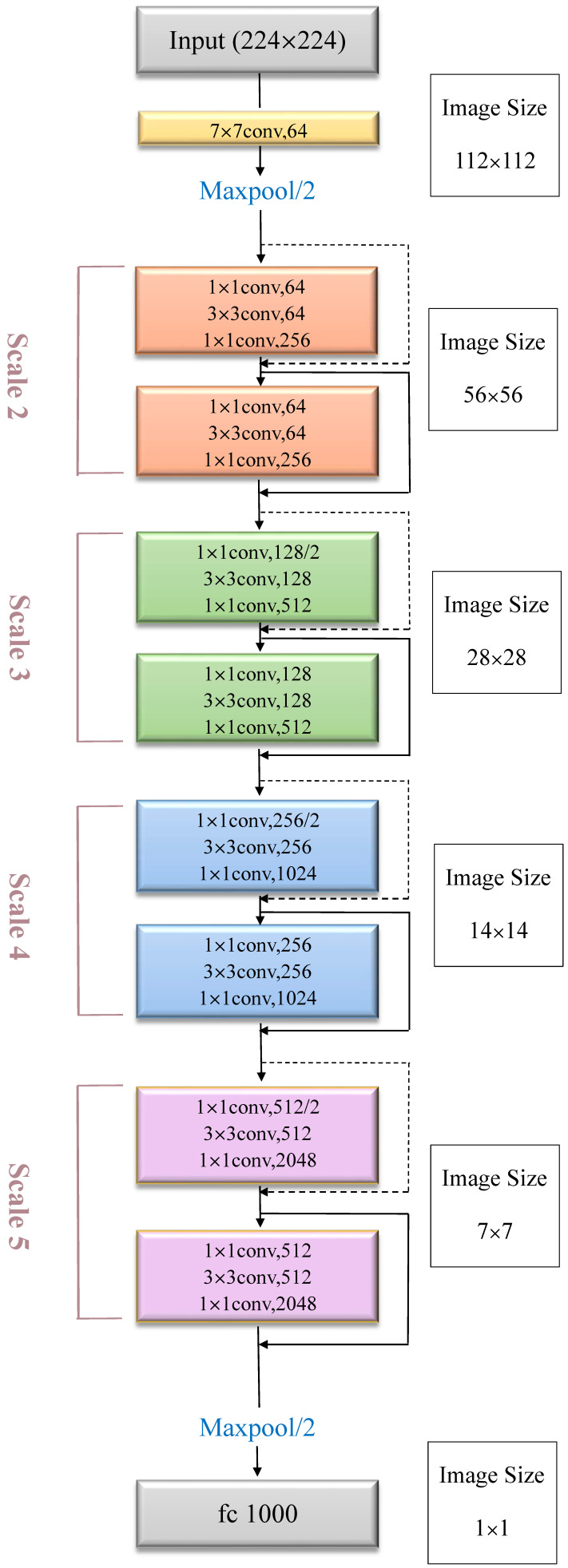
Resnet50 model for the detection of strawberry diseases.

**Figure 11 plants-10-00031-f011:**
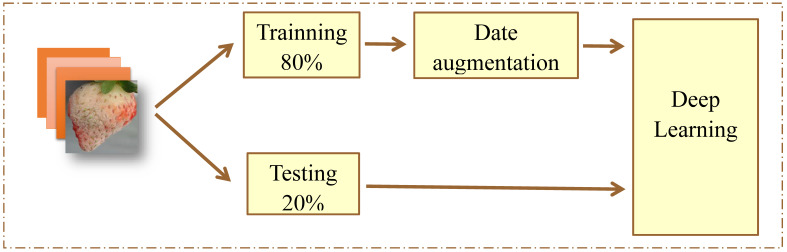
CNN training ratio distribution diagram for detection of strawberry diseases.

**Figure 12 plants-10-00031-f012:**
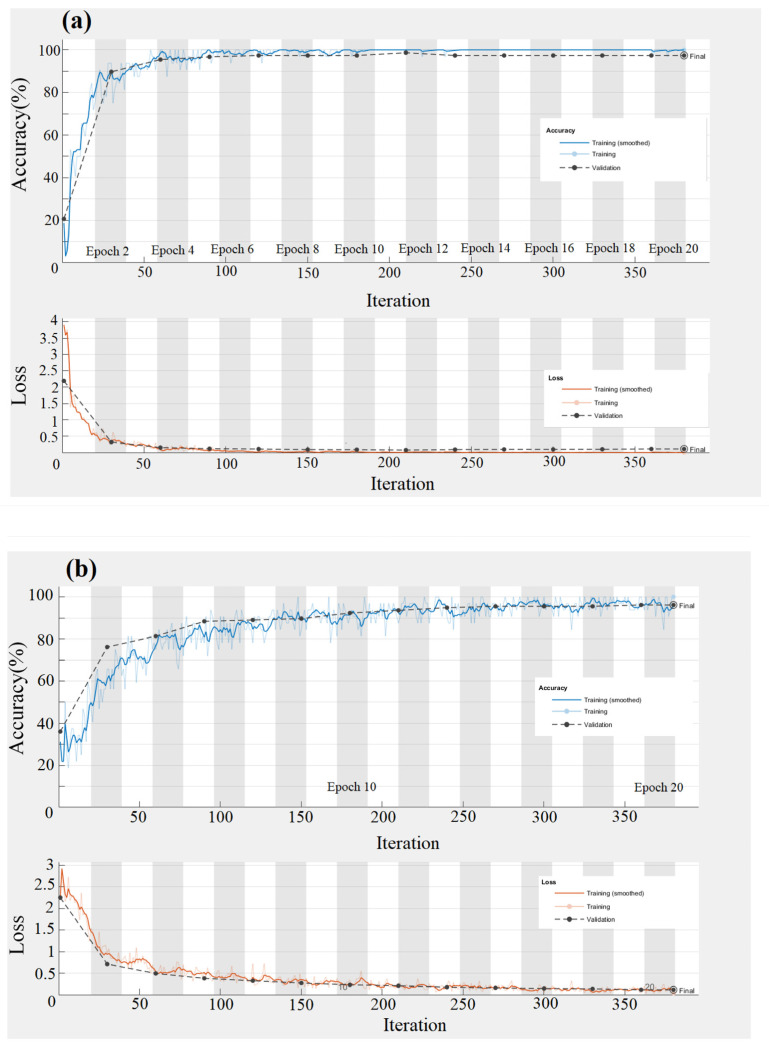
Accuracy and loss training map of original data set—(**a**) Vgg16, (**b**) GoogLeNet (**c**) Resnet50.

**Figure 13 plants-10-00031-f013:**
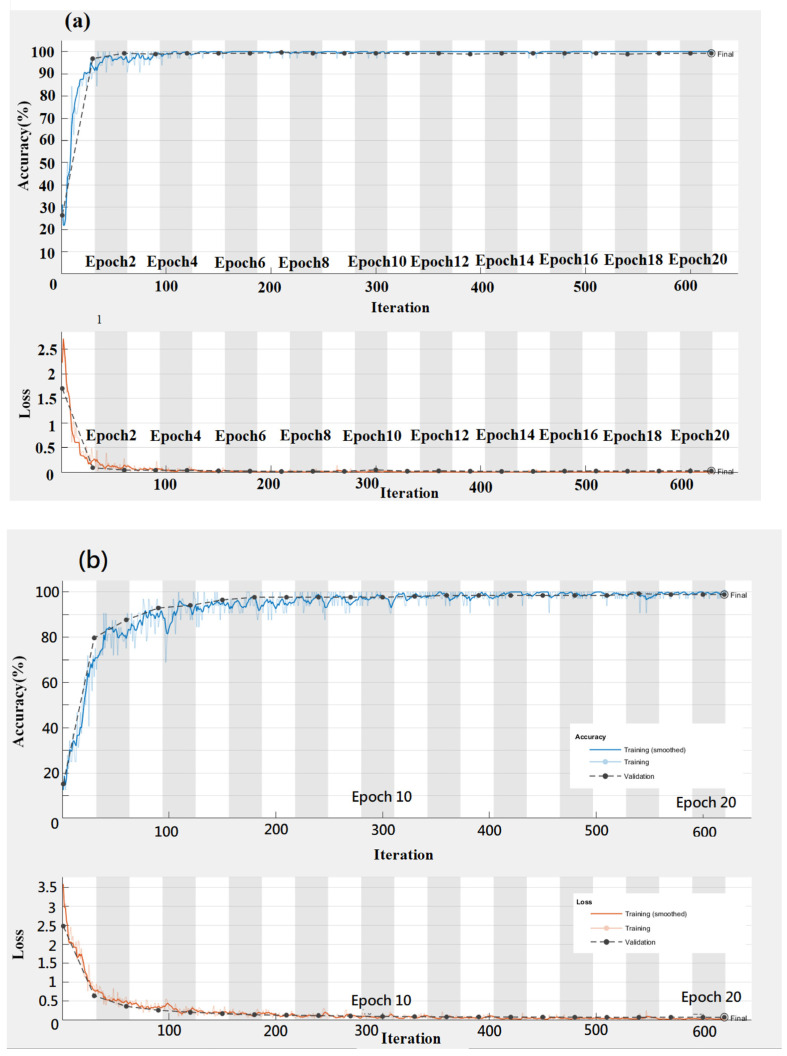
Accuracy and loss training map of feature data set—(**a**) VGG16, (**b**) GoogLeNet (**c**) Resnet50.

**Table 1 plants-10-00031-t001:** Table of numbers of disease images from original and feature data sets.

Diseases	Tissues	Original Images	Feature Images
leaf blight	crown	156	267
	leaf	166	262
	fruit	155	254
gray mold	fruit	157	250
powdery mildew	fruit	158	273
Total		792	1306

**Table 2 plants-10-00031-t002:** Model training parameters.

Parameter Name	Value
Optimization	sgd
Epochs	20
ValidationFrequency	30
Mini Batch size	32
Learning rate	0.0001
Execution environment	GPU

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
