# Peer review of "Detection of Strawberry Diseases Using a Convolutional Neural Network"

_plants, 2020, doi:10.3390/plants10010031_

Round 1
Reviewer 1 Report
Plants 1038419
The above paper examines the detection of diseases in strawberry plants using image analyses in Taiwan over a single season. The approaches used by the authors appear sound and the results are relatively novel for strawberry crops. The manuscript is well written and suitable for publication in Plants.
I include a few comments for the authors’ consideration.
It would good if the authors could include a statement that further research is required to compare the accuracy of image analysis with those of experienced crop scouts or other plant pathology specialists. I suspect that experienced operators are as good as the new system proposed in the paper. It would be interesting to compare yields in different plots using the two main disease reporting systems (digital versus experienced crop scout).
Can the authors provide data (yield losses) on the importance of the various leaf and fruit diseases for commercial strawberry production in Taiwan?
I found the order of the text confusing. I think it would help the reader if the ‘Methods’ was provided before the ‘Results and Discussion’.
Author Response
Reviewer #1
Reviewer’s comments:
The above paper examines the detection of diseases in strawberry plants using image analyses in Taiwan over a single season. The approaches used by the authors appear sound and the results are relatively novel for strawberry crops. The manuscript is well written and suitable for publication in Plants.
I include a few comments for the authors’ consideration.
It would good if the authors could include a statement that further research is required to compare the accuracy of image analysis with those of experienced crop scouts or other plant pathology specialists. I suspect that experienced operators are as good as the new system proposed in the paper. It would be interesting to compare yields in different plots using the two main disease reporting systems (digital versus experienced crop scout).
Response to the reviewer’s comment:
According to the reviewer’s comment, the authors have added the statement regarding the comparison of the accuracy of proposed techniques and experienced scop scouts.
In page 4, line 126-127
In the future, more research will be conducted to compare the accuracy of the results obtained by proposed technique with those obtained from experienced crop scout in the field.
Reviewer’s comments:
Can the authors provide data (yield losses) on the importance of the various leaf and fruit diseases for commercial strawberry production in Taiwan?
Response to the reviewer’s comment:
According to the reviewer’s comment, the authors added more data on the importance of the various leaf and fruit diseases for commercial strawberry production in Taiwan.
In page 1, line 18-20
Abstract:
The strawberry (Fragaria × ananassa Duch.) is a high-value crop with an annual cultivated area of ~500 ha in Taiwan. Over 90% of strawberry cultivation is in Miaoli County. Unfortunately, various diseases significantly decrease strawberry production. The leaf and fruit disease became an epidemic in 1986. From 2010 to 2016, anthracnose crown rot caused the loss of 30–40% of seedlings and~20% of plants after transplanting.
Reviewer’s comments:
I found the order of the text confusing. I think it would help the reader if the ‘Methods’ was provided before the ‘Results and Discussion’.
Response to the reviewer’s comment:
According to the reviewer’s comment, the authors would like to confirm that the order is arranged following the regulation of the journal.
Reviewer 2 Report
Using the latest artificial intelligence technology to develop new methods dealing with plant disease is apparently of great interest to farmers and researchers as well. Using a dataset consisting of strawberry plants infected by different diseases as well as healthy plants, the authors of this manuscript developed retrained new models to detect pests and diseases. The author used convolutional neural networks and deep transfer learning for models developing. Convolutional neural networks and deep transfer learning have been proved to be robust algorisms for image recognizing and classification. In my opinion, the authors picked up a set of right tools and developed useful new methods. However, there are still a major concerns to be addressed.
Introduction :
Introduction need much more clarity to describe the objective of the research, I would suggest to add clear objectives one by one end of the intro. Since authors used multiple plant parts to classify various diseases , author need to emphasis the importance of using multiple plant parts, see the below reference banana data sets please refer this paper
Selvaraj, M.G., Vergara, A., Ruiz, H. et al. AI-powered banana diseases and pest detection. Plant Methods 15, 92 (2019). https://doi.org/10.1186/s13007-019-0475-z
Materials and methods :
Table 1, what about health plant data sets ??? do you collect healthy classes ??
please add those data sets to enhance the value of this paper.
Results and discussion :
Results and discussion need more clarity, authors could be further extended the discussion how these results will help to develop a full automated AI mobile app to help millions of users.
Author Response
Reviewer #2
Reviewer’s comments:
Using the latest artificial intelligence technology to develop new methods dealing with plant disease is apparently of great interest to farmers and researchers as well. Using a dataset consisting of strawberry plants infected by different diseases as well as healthy plants, the authors of this manuscript developed retrained new models to detect pests and diseases. The author used convolutional neural networks and deep transfer learning for models developing. Convolutional neural networks and deep transfer learning have been proved to be robust algorisms for image recognizing and classification. In my opinion, the authors picked up a set of right tools and developed useful new methods. However, there are still a major concern to be addressed.
Introduction:
Introduction need much more clarity to describe the objective of the research, I would suggest to add clear objectives one by one end of the intro. Since authors used multiple plant parts to classify various diseases, author need to emphasis the importance of using multiple plant parts, see the below reference banana data sets please refer this paper
Selvaraj, M.G., Vergara, A., Ruiz, H. et al. AI-powered banana diseases and pest detection. Plant Methods 15, 92 (2019). https://doi.org/10.1186/s13007-019-0475-z
Response to the reviewer’s comment:
According to the reviewer’s comment, the authors have emphasized the objective of the research in the introduction section. Furthermore, we have added the importance of using five types of the disease. We also cite the paper Plant Methods 15, 92 (2019) into the manuscript.
In page 2, line 68-73
The objective of this study is to apply the state-of-the-art CNN techniques for the detection of leaf blight, gray mold, and powdery mildew from the “Taoyuan No. 1” and “Xiang-Shui” strawberry cultivars in Miaoli County, Taiwan. We also compare the accuracy of results obtained from three different models namely VGG-16, GoogLeNet, and Resnet-50 to confirm the feasibility of the proposed technique for strawberry disease detection.
In page 4, line 141-143
Images were taken for the “Taoyuan No. 1” and “Xiang-Shui” strawberry cultivars. These images were taken at a strawberry farm at Dahu Township, Miaoli County using a Sony RX10ii camera. During the shooting process, a plant pathologist identified the type of disease and marked them by nametags. Five types of strawberry disease images including leaf blight (crown rot, leaf blight, fruit rot), gray mold, and powdery mildew were collected. We choose these 5 types of disease for detection purpose because they are major diseases caused the yield lost for commercial strawberry production in Taiwan.
In page 2, line 58-60
In published reports, the convolution neural network (CNN) is one of the most popular ML techniques for detecting crop diseases. Jeon and Rhee [18] proposed the CNN technique for plant leaf recognition using the GoogLeNet model. The proposed technique was able to detect damaged leaves with a recognition rate of > 94%, even when only 30% of the leaf was damaged. Mohanty et al. [19] used CNN for detecting crop species and diseases based on a public dataset of images using GoogLeNet and AlexNet training models. Based on the color, grayscale, and leaf segmentation, the proposed model was 99.35% accurate. Selvaraj et al. [20] proposed deep transfer learning (DTL) for the detection of banana diseases and pest with the accuracy of 90%.
Reviewer’s comments:
Materials and methods:
Table 1, what about health plant data sets ??? do you collect healthy classes??
please add those data sets to enhance the value of this paper.
Response to the reviewer’s comment:
According to the reviewer’s comment, the authors would like to confirm that we did not collected the healthy data for the detection in the current manuscript. We acknowledge with thanks for this useful comment. We will consider the healthy data set in our future study to enhance the value of the paper.
In page 5, line 161-162
Each of the withered leaves was cut off as a feature image. The numbers of original images and feature images are shown in Table 1. Two hundred sixty-seven feature images were trimmed from 156 original images for crown rot caused by leaf blight disease, 262 feature images were trimmed from 166 original images for leaf blight diseases, 253 feature images were trimmed from 155 original images for fruit rot caused by leaf blight disease, 250 feature images were trimmed from 157 original images for gray mold diseases, and 273 feature images were trimmed from 158 original images for powdery mildew disease. A total of 1,306 images were trimmed from original images and named as feature images (Fig. 3). It is noted that the healthy dataset was not collected in this study. However, in the future, the healthy dataset will be used to enhance the impact of the proposed technique.
Reviewer’s comments:
Results and discussion:
Results and discussion need more clarity, authors could be further extended the discussion how these results will help to develop a full automated AI mobile app to help millions of users.
Response to the reviewer’s comment:
According to the reviewer’s comment, the authors have added more discussion in the “results and discussion” section. We also added more discussion on how the results will help to develop a full automated AI mobile app to help millions of users.
In page 4, line 117-131
The lower performance of the original image dataset can be explained by the complexity of the background pattern lead to the confusion of the system. Notably, this classification rate is comparable with that of the faster R-CNN multi-task learning proposed in [29-30] and higher than that of the supervised machine learning in [21,24]. In general, the proposed technique provides a potential method for five major strawberry diseases detection. The classification rates obtained from feature image dataset is better than those obtained from original image dataset with all three models. The fruit leaf blight and leaf blight diseases are detected with 100% classification rate with all three models. While the crown leaf blight disease is detected with 100% classification rate with only two models of VGG16 and Resnet50. The gray mold disease is detected with 100% classification rate with only VGG16 model. The powdery disease is detected with 100% classification rate with only GoogLeNet model. In the future, more research will be conducted to compare the accuracy of the results obtained by proposed technique with those obtained from experienced crop scout in the field. Furthermore, based on the fruitful results obtained from the proposed technique, a full automated artificial intelligent mobile app will be developed to perform disease detection in early stage. Thus, it will control the yield losses for commercial Strawberry production in Taiwan and help millions of users in the developed countries.
Reviewer 3 Report
The work is interesting and valuable. It concerns the use of the CNN method to identify diseases in strawberries. The authors emphasize that such a method has not been used in this species so far, and the conducted research confirms its usefulness. However, the manuscript requires little correction before publication.
The introduction satisfactorily introduces the issues raised at work. The division of material and method is described in great detail and clearly. The chapter results and discussion require editing. This chapter is very short. The results are described clearly and concisely and confirm the possibility of using the CNN method to identify diseases in strawberries. However, this chapter lacks a discussion with the works of other authors, a comparison of the results and their reference to previously published works in this field.
Author Response
Reviewer #3
Reviewer’s comments:
The work is interesting and valuable. It concerns the use of the CNN method to identify diseases in strawberries. The authors emphasize that such a method has not been used in this species so far, and the conducted research confirms its usefulness. However, the manuscript requires little correction before publication.
The introduction satisfactorily introduces the issues raised at work. The division of material and method is described in great detail and clearly. The chapter results and discussion require editing. This chapter is very short. The results are described clearly and concisely and confirm the possibility of using the CNN method to identify diseases in strawberries. However, this chapter lacks a discussion with the works of other authors, a comparison of the results and their reference to previously published works in this field.
Response to the reviewer’s comment:
According to the reviewer’s comment, the authors have added more discussion and comparison with previously published works in this field.
In page 4, line 117-119
The lower performance of the original image dataset can be explained by the complexity of the background pattern lead to the confusion of the system. Notably, this classification rate is comparable with that of the faster R-CNN multi-task learning proposed in [29-30] and higher than that of the supervised machine learning in [21,24].
Reference
- Gao, Z. M., Shao, Y. Y., Xuan, G. T., Wang, Y. X., Liu, Y., and Han, X. (2020). Realtime hyperspectral imaging for the in-field estimation of strawberry ripeness with deep learning, Intell. Agri 4, 31-38. https://doi.org/10.1016/j.aiia.2020.04.003
- Nie, X., Wang, L., Ding, H. X., and Xu, M. (2020). Strawberry verticillium wilt detection network based on multi-task learning and attention, IEEE Acccess 7, 170004. DOI: 1109/ACCESS.2019.2954845
Round 2
Reviewer 2 Report
Thanks for the response and corrections.